# The Influence of Negative Voltage on Corrosion Behavior of Ceramic Coatings Prepared by MAO Treatment on Steel

Mingzhe Xiang [1], Tianlu Li [1], Yun Zhao [1,2,3,*] and Minfang Chen [1,2,3,*]

1   School of Materials Science and Engineering, Tianjin University of Technology, Tianjin 300384, China;
    xmz120002@163.com (M.X.); tjlitianlu@126.com (T.L.)
2   Key Laboratory of Display Materials and Photoelectric Device (Ministry of Education),
    Tianjin University of Technology, Tianjin 300384, China
3   National Demonstration Center for Experimental Function Materials Education,
    Tianjin University of Technology, Tianjin 300384, China
*   Correspondence: yun_zhaotju@163.com (Y.Z.); mfchentj@126.com (M.C.)

**Abstract:** In this study, the 10B21 steel was treated by micro-arc oxidation (MAO) in the electrolyte consisting of aluminate and phosphate to enhance its corrosion resistance. The effect of negative voltage on corrosion resistance of the MAO coating obtained has mainly been analyzed through their phase composition, microstructure, interfacial bonding strength, salt spray and electrochemical testing. The result indicates that with negative voltages applied to the MAO coating, Fe-Al transition layers arise between coating layer and matrix. Furthermore, different negative voltages bring forward different $\alpha$-$Al_2O_3$ contents contained in the MAO coatings and when it reaches $-100$ V, the amount of $\alpha$-$Al_2O_3$ appears as the largest. The surface porosity of the coating was also significantly decreased. In addition, the corrosion current density of the coating is only 3.64 $\mu A \cdot cm^{-2}$, which is two orders of magnitude lower than that of the substrate. After 72 h of salt spray corrosion, it is found that the coating substrate is less corroded when the negative voltage of 100 V is applied. Therefore, $-100$ V has been proven as the optimum performance for improving the corrosion resistance of 10B21 steel.

**Keywords:** 10B21 steel; microarc oxidation; negative voltage; corrosion resistance

## 1. Introduction

Thin-wall integrated parts of magnesium alloy vehicles usually have complex geometrical shapes connected with car bodies by punch riveting. The rivets used for this connection are manufactured from low carbon steel 10B21 containing B. The direct contact between steel rivet and magnesium alloy parts usually results in significant galvanic corrosion due to their potential difference of up to 0.934 V [1]. Surface treatment is an effective way to reduce the potential difference, especially for rivets, being simpler and easier. The traditional approaches for coated rivets include hot-dip plating and blackening [2–6]. Nonetheless, the coating cannot be effectively insulated because of process complexity, coating defects and weak adhesion, especially for a long time. At present, galvanized rivets are usually utilized for this cold connection in the automobile industry. Zinc coating can effectively reduce the potential difference between steel rivets and magnesium alloy parts, and their galvanic corrosion is not severe. However, the corrosion current density of galvanized steel is $1.51 \times 10^{-3}$ $A \cdot cm^{-2}$ [7], and there still exists a great risk of corrosion for joints, which reduces the safety of magnesium alloy components.

Microarc oxidation, as a protection technique, has been greatly applied for fabricating hard ceramic coatings on the Mg, Al, Ti, and other valve metals and their alloys through instantaneous arc discharge reactions [8–13]. Steel materials can be easily oxidized to $Fe^{3+}$, and the preparation of MAO ceramic coatings usually has particular requirements for electrolytes. Pezzato et al. prepared an in situ hard ceramic coating that consisted mainly of $Fe_2O_3$ on the surface of 39NiCrMo3 alloy steel in an electrolytic liquid system with

$Na_2SiO_3$ as the main component and NaOH and $NaAlO_2$ as the additional components [14]. However, $Fe_2O_3$ would lead to galvanic corrosion to accelerate the corrosion of the substrate. Wang et al. [15–18] added $Na_2WO_4$ and $Na_2SO_4$ into the $Na_2SiO_3$ electrolyte to optimize the electrolyte composition, which was favorable for forming arcs on Q235 steel; however, their corrosion resistances were not discussed. In addition, composed mainly of $\alpha$-$Al_2O_3$ and $\gamma$-$Al_2O_3$, hard-ceramic coatings on low carbon steel have been reported [19] as having good hardness and friction properties due to the excellent comprehensive properties of $\alpha$-$Al_2O_3$. However, with plenty of holes and cracks, the coating was thin, reaching approximately 25 μm. Li also reported that a ceramic coating consisted mainly of $\alpha$-$Al_2O_3$ was formed on the surface of 10B21 steel in $NaAlO_2$ and $NaH_2PO_4$ electrolytes optimized by $Na_2B_4O_7$ and $Na_2CO_3$ [20]. The thickness of the coating reached approximately 100 μm after 30 min MAO operation. Generally, the thickness of coating used for industrial application is approximately 30 to 50 μm, while the coating with excessive thickness has no prominent significance to improving the corrosion resistance of substrate. Moreover, overlong time of processing would also reduce production efficiency and increase industry cost. In addition, we prepared MAO films by adding $\alpha$-$Al_2O_3$ into the MAO electrolyte solution as an additional component to 10B21 [21], and it was found in our study that the holes in the coating can be filled, and the performance of the films can be enhanced. However, with the potential possibility that $\alpha$-$Al_2O_3$ agglomerates on the surface, the corrosion current density is only 0.2 A·cm$^{-2}$. Therefore, further increasing the $\alpha$-$Al_2O_3$ content and evenly distributing it in the coating of 10B21 steel is crucial for the riveting of automobile stamping. In this study, NaOH is added directly to the electrolyte to accelerate the hydrolysis of $NaAlO_2$ to produce a large amount of homogeneous $\alpha$-$Al_2O_3$.

Generally, the electrical parameters [22–24] conducted in micro-arc oxidation have significantly influenced the coating performance. As a critical parameter, a negative voltage used as a coating melting voltage can improve the crystallinity of the ceramic layer, influence the wear resistance, and contribute to the growth efficiency of the ceramic coating. In particular, the discharge process melts the loose layer of micro-arc oxidation, thereby reducing the pores and defects of the MAO coating. Di et al. [25] found that an adequate raise in the positive and negative pulse ratios reduced the coating roughness of AZ91D magnesium alloy. In addition, the role of the negative voltage in the MAO process of AZ91D is to restrain the large arc tendency for uniformly distributed arc point and a smooth film surface [26]. Li et al. [27] revealed that the wear resistance of 2A50 aluminum alloy was improved by a negative voltage, which affected the amount and size of micropores formed in the MAO operation. The hardness of the film under $-100$ V was the highest with stable friction coefficient. Although many studies have focused on the influence of negative voltage on the composition, micromorphology, and properties of MAO coatings, the substrates used for MAO operation have always been magnesium, aluminum, titanium, etc. There are few works concerning MAO operation of non-valve metals such as steel; particularly, problems such as the effect of negative voltage on the MAO coating and its growth should be further explored.

Based on previous research, the present study aims to prepare $\alpha$-$Al_2O_3$-containing ceramic coatings on the surface of 10B21 through the MAO operation under different negative voltages in aluminate and phosphate systems, to achieve which, the influence of negative voltages on the composition, microstructure, bonding strength, and corrosion performance of these MAO coating will be systemically investigated as well as the corresponding mechanism of coating growth under different negative voltages on 10B21 steel.

## 2. Materials and Methods

### 2.1. Materials

The sample substrate was a piece of 10B21 steel ($\Phi$30 mm $\times$ 2 mm) that was ultrasonically cleaned in acetone and ethyl alcohol and dried with cold air later after it was ground to 1500 grits using SiC paper. Its chemical composition is listed in Table 1.

**Table 1.** The composition of 10B21 steel (wt.%).

| Element | C | Si | Mn | P | S | B |
|---------|---|-----|------|------|------|------|
| Content | 0.18–0.23 | ≤0.10 | 0.70–1.00 | ≤0.030 | ≤0.035 | ≥0.0008 |

### 2.2. Preparation of MAO Coatings

To prepare the MAO coatings, the present research deployed the research sample as the MAO (MAO-50D pulse power source from Tongchuang Technology Co., Ltd., Chengdu, China) anode, and the stainless-steel plate as the cathode inside an electrolyte containing 15 g/L $NaAlO_2$, 3 g/L $NaH_2PO_4$, 3 g/L $Na_2B_4O_7$, and 0.5 g/L NaOH (electrolyte temperature below 40 °C), and set the positive voltage to 500 V, the frequency to 1200 Hz, the duty ratio to 30%, and the negative voltage as the variate, as shown in Table 2. Then the MAO test was run for 20 min.

**Table 2.** The experimental groups.

| F1 | F2 | F3 | F4 | F5 |
|------|-------|--------|--------|--------|
| 0 V | −50 V | −100 V | −150 V | −200 V |

### 2.3. Characterization of MAO Coatings

In the present research, a scanning electron microscope (SEM, Quanta FEG 450, Ann Arbor, MI, USA) and an EDS spectrometer were used to investigate the microstructures and elemental distributions of the sample with surface porosity analyzed by ImageJ v1.37 software, X-ray diffraction (XRD, D-max2500, Tokyo, Japan) was employed to test the sample composition with the Cu K$\alpha$ radiation ranging from $10°$ to $80°$ at a speed of $8°$ $min^{-1}$ with results analyzed by JADE6 software, and X-ray photoelectron spectroscopy (XPS, ESCALAB250Xi, Thermo Fisher Scientific, Waltham, MA, USA) with a monochromatic Al K$\alpha$ radiation ($\lambda$ = 8.4 nm) was utilized to analyze the sample space elements with the results fitted by the XPSPEAK4.1 software.

### 2.4. Corrosion Characterization

Electrochemical testing was carried out by three-electrode system from a Zennium electrochemical workstation in a 3.5 wt.% NaCl solution. A sample of 1 $cm^2$ was taken as the working electrode with a saturated calomel electrode as the reference electrode, and the auxiliary electrode as graphite. The samples were immersed for 30 min in NaCl solution before the electrochemical testing that had a 1 mV/s scan rate in a frequency range $10^{-2}$–$10^5$ Hz. The polarization resistance was calculated by the following formula (Equation (1) [28]). $R_p$: polarization resistance; $\beta_a$: Tafel potential of the anode; $\beta_c$: Tafel potential of the cathode; $I_{corr}$: corrosion current density.

$$R_p = \frac{\beta_a \beta_c}{2.303(\beta_a + \beta_c)I_{corr}} \tag{1}$$

The Neutral salt spray (NSS) test was run for different durations in its test chamber (YWX-150), defined according to QB/T 382999. After each preset time was due, the testing sample was taken out of the chamber. Then, the sample was weighed after removing the corrosion products on its surface by the solution prepared according to GB/T 1654996. The corrosion rate is calculated according to Equation (2) [20].

$$V = \frac{W_0 - W_1}{S \times T} \tag{2}$$

$V$: weight loss in corrosion rate (g·$m^{-2}$·$h^{-1}$); $W_0$: sample weight before salt spray test (g); $W_1$: sample weight after salt spray test (g); $S$: corrosion area ($m^2$); $T$: corrosion time (h).

*2.5. Bond Strength Test*

A WS-2005 automatic coating adhesion scratcher (indenter diameter 0.4 mm) was used to test the coating bonding force under 40 N load strength and 3 mm scratch length. Each sample was tested in three different areas to obtain an average value. The bonding strength was obtained by Equation (3) [21].

$$P = \frac{F}{S} \tag{3}$$

*P* is the bonding strength of the coating (MPa); *F* is the load force corresponding to the first signal peak (N); *S* is the contact area between the pressure head and the coating (m$^2$).

## 3. Results and Discussions

### 3.1. Coating Composition

The XRD patterns of the obtained samples are displayed in Figure 1a. The MAO coating is composed mainly of $\alpha$-Al$_2$O$_3$, $\gamma$-Al$_2$O$_3$, Fe$_3$O$_4$, FeAl$_2$O$_4$, and FePO$_4$, and an intense diffraction peak corresponding to Fe can be easily observed because the X-rays can penetrate the coating to the substrate due to the porous structure of the coating. Meanwhile, all samples display diffraction peaks of Fe$_3$O$_4$ and FePO$_4$ at 45.5° and 65.0°, respectively, indicating that the 10B21 steel substrate participates in the MAO reaction [29,30]. Usually, the diffraction peak corresponding to FeAl$_2$O$_4$ is at 45.5°, which overlaps with that of Fe$_3$O$_4$ [16]. The diffraction peaks ascribed to $\alpha$-Al$_2$O$_3$ are at 2$\theta$ values of 31.7°, 37.2°, 39.1°, 60.3°, and 66.5°, and the diffraction peaks at 43.3° and 56.3° correspond to $\gamma$-Al$_2$O$_3$. Generally, different crystal types of alumina present in the coating can directly affect the corrosion resistance of specimens [19,31]. Therefore, the peak strength ratio (I$_{\alpha/\gamma}$) of the maximum diffraction peak of $\alpha$-Al$_2$O$_3$ at 37.2° and $\gamma$-Al$_2$O$_3$ at 43.3° is calculated for each coating according to [21], and the results are shown in Figure 1b. Specifically, the F1 coating has the lowest value of I$_{\alpha/\gamma}$, indicating the higher content of $\gamma$-Al$_2$O$_3$ with low crystallinity in its coating. With a negative voltage increasing from F2 to F5, the I$_{\alpha/\gamma}$ value rises first and then decreases, and the I$_{\alpha/\gamma}$ of F3 is higher than 7.0. Due to the lower negative voltage leading to the slower hydrolysis reaction of aluminum hydroxide, the F2 coating probably still has a relatively lower content of $\alpha$-Al$_2$O$_3$. Under the preparation conditions of F3, a further increment of current density leads to a significant increase in the reaction temperature, and this significantly accelerates the transformation of $\gamma$-Al$_2$O$_3$ with low crystallinity to $\alpha$-Al$_2$O$_3$ with high crystallinity, thus resulting in the maximum content of $\alpha$-Al$_2$O$_3$. Additionally, there is an apparent decline in I$_{\alpha/\gamma}$ in the coatings of F4 and F5, which is probably due to the higher pulse energy from excessive negative voltage leading to the partial dissolution of formed $\alpha$-Al$_2$O$_3$.

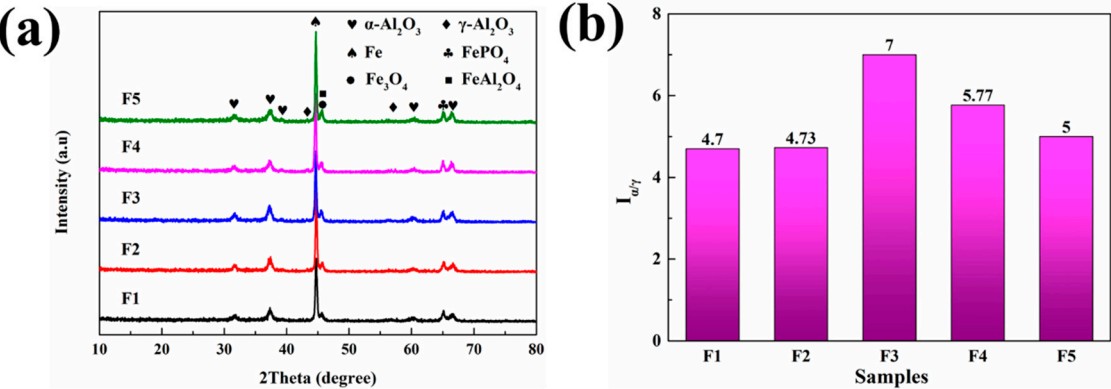

**Figure 1.** (**a**) The XRD patterns of samples; (**b**) I$_{\alpha/\gamma}$ of samples under different negative voltages.

*3.2. Surface Morphology Analysis*

Figure 2a–e displays the SEM morphology of the coating surface under different negative voltages. The surface porosity of the coating by the statistics of ImageJ software is also shown in Figure 2f. The sample surface under each negative voltage has the volcanic hole characteristic [32]. The papillary-like morphology (point A in Figure 2a) observed in the high-magnification image is the main surface feature in the F1 and F2 coatings, and the formation process of this type of morphology is known as an A-type discharge event [33,34]. Briefly, under a critical value of negative voltage, the released energy is comparatively less, and A-type discharge occurring in the outer layer of the coating produces a porous surface that consists mainly of $\gamma$-Al$_2$O$_3$ from NaAlO$_2$ hydrolysis reaction in electrolyte. Discharge energy is further increased as the increasing of the negative voltage, and the electric current brings more Al$^{3+}$ into the discharge channel formed by H$_2$ release from the cathode [35]. This process produces a large amount of $\gamma$-Al$_2$O$_3$ at the interface between substrate and coating, and then $\gamma$-Al$_2$O$_3$ converts into molten $\alpha$-Al$_2$O$_3$ and is ejected again through the discharge channel under high temperature. Furthermore, a pancake-like morphology (point B in Figure 2c) from $\alpha$-Al$_2$O$_3$ solidification and stacking is obtained, illustrating that the B-type discharge event occurs in the MAO process, which is conducive to obtaining a compact surface, and plays a dominant role for coating growth.

In F1, there is almost no cathode current on the surface. Therefore, A-type discharge events dominate the surface reaction, and the porosity of the papillary-like morphology is 10.7%. With the addition of a negative voltage, although a weak current produced by a negative voltage cannot cause B-type discharge, its porosity still decreases to 9.1%. When B-type discharge occurs, the negative voltage is increased to −100 V. Under this condition, both A and B discharges are carried out simultaneously on the sample surface, and B-type discharge dominates the reaction process. The porosity of the pancake-like morphology, flat and compact, further declines to 7.4%. Nonetheless, the excessive discharge energy provided by the overhigh negative voltage in F4 and F5 leads to an increase in porosity, in which the solidified alumina of the formed coating is melted again and thus many cracks and deep holes are produced, despite B-type discharge still occurring.

The EDS results in Figure 2(a1–e1) and Table 3 show that the content of Al in the coating varies greatly from F1 to F5, while the variation of the content of Fe is opposite, also implying that negative voltage significantly affects the composition content of MAO coating. The F3 coating with few micropores (Figure 2(c1)) has the highest Al content (73%) and the lowest Fe content (4%) among all samples, probably attributable to the promotion of the hydrolysis reaction of NaAlO$_2$ by negative voltage, which produces the deposition of more Al onto the coating surface.

**Table 3.** Element content of the MAO coating under different negative voltages.

| Sample | Al (%) | O (%) | Fe (%) | P (%) |
|--------|--------|-------|--------|-------|
| F1 | 63 | 24 | 5 | 8 |
| F2 | 67 | 20 | 7 | 6 |
| F3 | 73 | 17 | 4 | 6 |
| F4 | 72 | 18 | 5 | 6 |
| F5 | 67 | 19 | 10 | 4 |

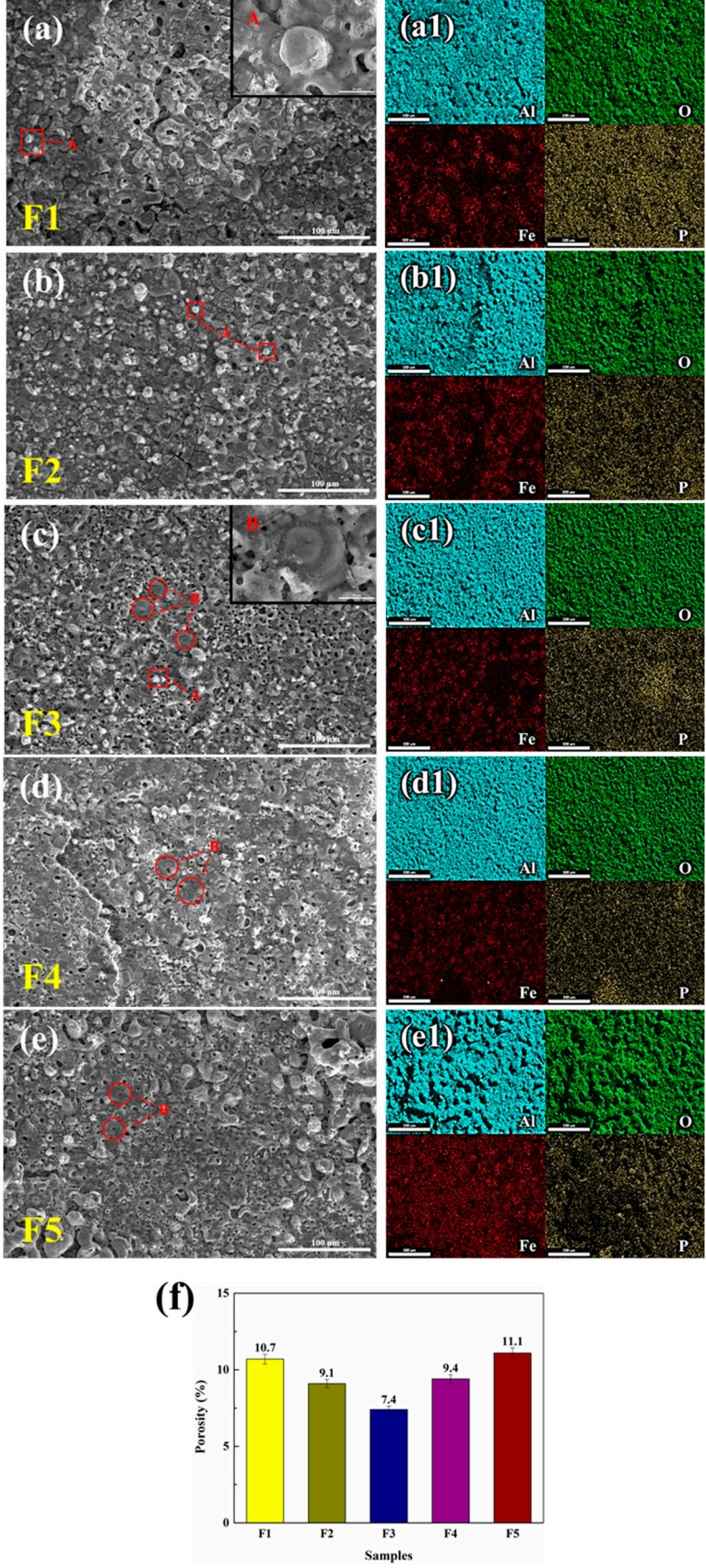

**Figure 2.** SEM images (1000×) of the coating surface and EDS analysis (2000×): (**a**,**a1**) F1; (**b**,**b1**) F2; (**c**,**c1**) F3; (**d**,**d1**) F4; (**e**,**e1**) F5 and (**f**) surface porosity of MAO coatings under different negative voltages.

### 3.3. Cross-Section Morphology Analysis

For a cross-section of the samples, there are apparent holes with different sizes observed in the outer coating in Figure 3a, and the F1 thickness is approximately 22 μm. The content variation of Al and O elements in the depth direction of the coating is relatively remarkable, indicating the porous and uncompacted structure present here, which is probably due to the composition of $\gamma$-$Al_2O_3$. There are also sudden increases and decreases in the Al and Fe element curves at the interface, which may be attributed to the absence of a $FeAl_2O_4$ transition layer [15]. In F2, the coating thickness is similar to the coating thickness of F1, and apparent cracks are observed at the sample interface, which may be caused by the insufficient reaction between Fe and Al in the transition layer under low negative voltage. With increasing negative voltage, the obvious cracks and deep holes in F3 (Figure 3c) disappear and the thickness of the coating increases significantly, reaching 34 μm. The content of Al and O elements is relatively constant in the thickness range of 10–25 μm across the coating region, indicating a uniform density and compact structure, possibly owing to the high $\alpha$-$Al_2O_3$ content in F3. The variation in the content curves of Al and Fe elements decreases and rises slowly at the interface between coating and substrate, implying that integrated $FeAl_2O_4$ exists on the transition layer, which might be caused by the appropriate negative voltage set. The difference in the thicknesses of the F3 and F4 coatings (Figure 3d) is slight, but large cracks are observed in the transition layer of F4, resulting in a loose and porous coating. The thickness of the F5 coating (Figure 3e) further decreases to 16 μm, albeit with the prominent transition layer. There are also more cracks and holes in the loose layer and the transition layer, which proves the disadvantages of an overhigh negative voltage to transition layer formation and coating quality.

### 3.4. XPS Analysis

The surface compositions of the F3 coating are further investigated by XPS analysis. The spectrum of the F3 is displayed in Figure 4. It reveals that the elements on the surface coating are mainly composed of O, Al, Fe, and Na, while the C present on the surface layer as shown in the spectrum is probably ascribed to the ambient contamination [36]. Meanwhile, the fitting peaks are found at 74.1 and 74.3 eV in the high resolution of Al2p spectra, ascribed to $\gamma$-$Al_2O_3$ and $\alpha$-$Al_2O_3$, respectively. The proportions of corresponding peaks areas of $\gamma$-$Al_2O_3$ and $\alpha$-$Al_2O_3$ are also calculated, as 11.9% and 88.1%, respectively, revealing the primary composition of $\alpha$-$Al_2O_3$ in the coating surface. Moreover, in the high-resolution spectra of Fe, it is shown that Fe2p3/2 has two characteristic peaks (711.0 and 711.4 eV), corresponding to $Fe^{2+}$ in $FeAl_2O_4$ and $Fe^{3+}$ in $Fe_3O_4$ respectively, as the former, which is produced by the reaction between FeO and $Al_2O_3$ in the coating, can further improve the corrosion resistance of the coating. Furthermore, after being fitted, the O1s spectrum (Figure 4d) performs two characteristic peaks at 531.0 and 531.6 eV. Specifically, the former is consistent with Fe and Al oxides in the coating, and the latter is Fe and Al hydroxides in the electrolyte [14,30]. In addition, it is shown in Table 4 that the Al content in the coating surface is significantly greater than that of Fe, implying that the outermost surface mainly consists of $Al_2O_3$ and $FeAl_2O_4$ [14].

**Table 4.** Quantitative analysis of the sample F3 coating.

| Element Region | C 1s | O 1s | Al 2p | Fe 2p |
| --- | --- | --- | --- | --- |
| Atomic (%) | 50.25 | 35.09 | 12.05 | 2.61 |

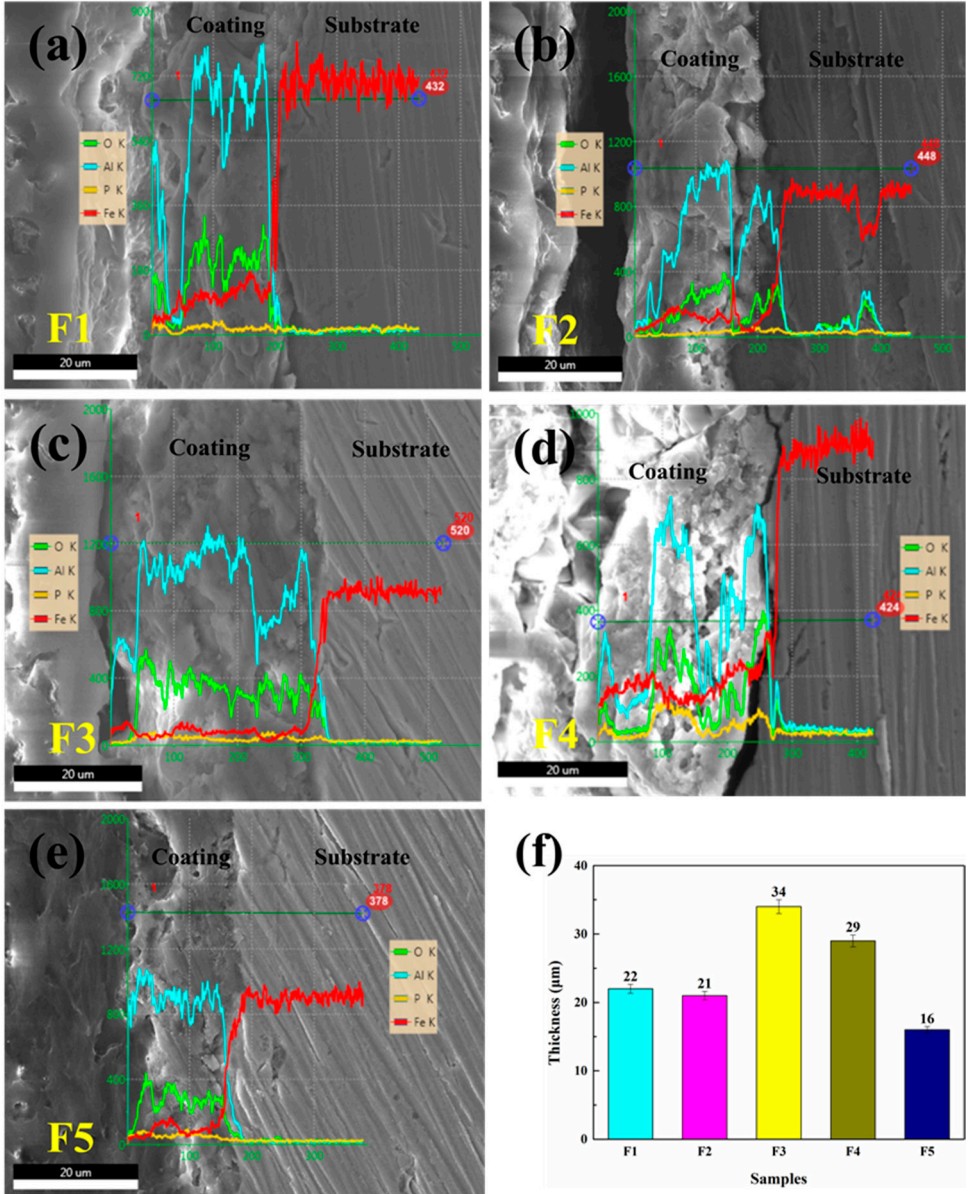

**Figure 3.** Cross section SEM images (5000×) of MAO coatings and EDS line scan distribution curve: (**a**) F1; (**b**) F2; (**c**) F3; (**d**) F4; (**e**) F5, and (**f**) the thickness of the coatings under different negative voltages.

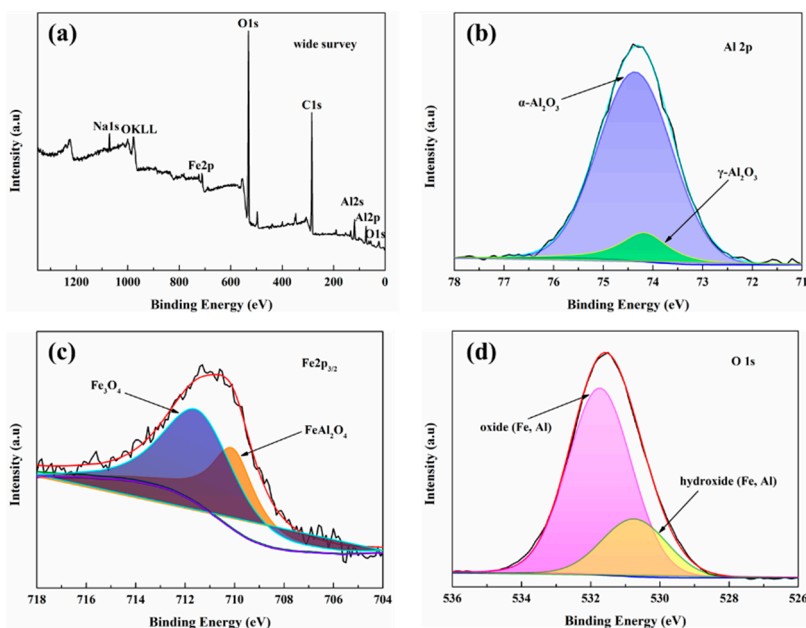

**Figure 4.** XPS spectra of the sample F3 coating: (**a**) Wide survey; (**b**) Al 2p; (**c**) Fe $2p_{3/2}$; (**d**) O 1s.

### 3.5. Formation Mechanism

With the addition of negative voltage, two significant changes are found in the coating: (1) the surface pores of the coating are reduced and a large amount of $\gamma$-$Al_2O_3$ is converted $\alpha$-$Al_2O_3$; (2) a stable Fe-Al transition zone is formed between the coating and the matrix. Figure 5 illustrates the formation mechanism of MAO coating before and after negative voltage application.

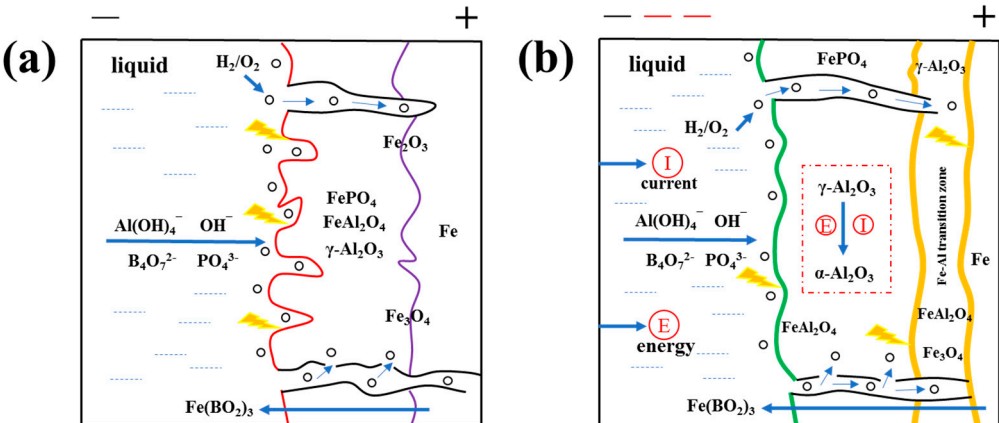

**Figure 5.** Schematic diagram of the MAO coating formation process before and after applying a negative voltage: (**a**) before and (**b**) after.

In the early stage of the MAO procedure, arc discharge produces much heat, leading to a rapid rise in electrolyte temperature and promoting the chemical reaction of $NaH_2PO_4$ in the electrolyte. Moreover, NaOH accelerates the hydrolyzation of $NaAlO_2$ [37]. Simultaneously, the water in the electrolyte is vaporized rapidly and bubbles are formed on substrate surface, making a channel for arc discharge and charge transfer. The formula for $Al_2O_3$ production during the MAO procedure is shown in Equations (4)–(7).

$$4OH^- - 4e^- \rightarrow O_2 \uparrow + 2H_2O \tag{4}$$

$$AlO_2^- + 2H_2O \rightarrow Al(OH)_3 + OH^- \tag{5}$$

$$2Al(OH)_3 \rightarrow Al_2O_3 + 3H_2O \tag{6}$$

$$H_2PO_4^- + 2Al(OH)_4^- \rightarrow PO_4^{3-} + Al_2O_3 + 5H_2O \tag{7}$$

At the same time, the Fe element from the substrate is easily oxidized into $Fe^{2+}$ and $Fe^{3+}$ because of the high-temperature discharge. Partial $Fe^{3+}$ and $PO_4{}^{3-}$ are formed into $FePO_4$ antirust, and FeO reacts with $Fe_2O_3$ and $Al_2O_3$ to turn into $Fe_3O_4$ and $FeAl_2O_4$, respectively, which improve the corrosion resistance of the coating. However, some $Fe^{2+}$ still reduce the conductivity of the substrate and inhibit the formation of $Al_2O_3$ on the coating. The additional component of $Na_2B_4O_7$ can be decomposed and then generate the glassy molten salt $B_2O_3$ under high temperature, which can effectively dissolve the harmful $Fe_2O_3$, becoming a favorable factor for the formation of $\alpha$-$Al_2O_3$. In addition, when the suitable negative voltage is applied, the Fe-Al transition layer appears, the layer contains $FeAl_2O_4$ which can improve the adhesion and corrosion resistance of the coating. The main chemical reactions are as follows (Equations (8)–(12)):

$$FeO + Fe_2O_3 \rightarrow Fe_3O_4 \tag{8}$$

$$Fe^{3+} + PO_4^{3-} \rightarrow FePO_4 \tag{9}$$

$$Na_2B_4O_7 \overset{\Delta}{\rightarrow} 2NaBO_2 + B_2O_3 \tag{10}$$

$$Fe_2O_3 + 2B_2O_3 \rightarrow Fe(BO_2)_3 + FeBO_3 \tag{11}$$

$$FeO + Al_2O_3 \rightarrow FeAl_2O_4 \tag{12}$$

Generally, an inappropriate negative voltage with insufficient or excessive energy would lead to uneven growth or inevitable destruction of the Fe-Al transition layer. Meanwhile, negative voltage has been found to influence the content of coating composition, especially that of $\alpha$-$Al_2O_3$. The amount of $\alpha$-$Al_2O_3$ generated on the coating surface is increased, which is favorable to enhance the corrosion resistance of alloy substrate [32]. The instantaneous temperature generated by a negative voltage is a crucial factor for the crystal transition of $Al_2O_3$ (Equation (13)) [33] and the reduction of coating cracks and holes through the melting and solidification process. Undoubtedly, coating thickness also varies during this process. The change of negative voltage affects the content of $\alpha$-$Al_2O_3$ in the coating. When the negative voltage is 100 V, the content is the highest. Therefore, the surface of F3 coating has the highest density and thickness, and the number of holes is decreased. In addition, the coating may also perform the best corrosion resistance [38].

$$\gamma - Al_2O_3 \overset{\Delta}{\rightarrow} \alpha - Al_2O_3 \tag{13}$$

### 3.6. Coating Bonding Strength

The first acoustic signal peak position corresponds to the amount of load, that is, the bonding force in Figure 6. Table 5 displays the bonding strength of the five coatings. The F1 coating without a negative voltage has the lowest bonding strength because the coating has no prominent Fe-Al transition layer, while having too much $Fe_2O_3$, $Fe_3O_4$ and $\gamma$-$Al_2O_3$ on the interface. When a negative voltage is applied, the bonding strength of the F2 coating increases prominently on account of the appearance of the Fe-Al transition layer, which is beneficial to preventing the further diffusion of iron oxide. The bonding strength of the F3 coating is the highest, reaching $118.2 \pm 3.54$ MPa, showing the best bonding performance. When the proper negative voltage is applied, the loose layer of $\gamma$-$Al_2O_3$ almost disappears, and the thickness of the $\alpha$-$Al_2O_3$ dense layer increases. Moreover, the Fe-Al transition layer is more stable, which significantly improves the bonding force of the coating. With further negative voltage increased, the F4 and F5 bonding strengths are found to decline because of the destruction of the $\alpha$-$Al_2O_3$ layer and Fe-Al transition layer. Therefore, the bonding performance of the corresponding coatings decline.

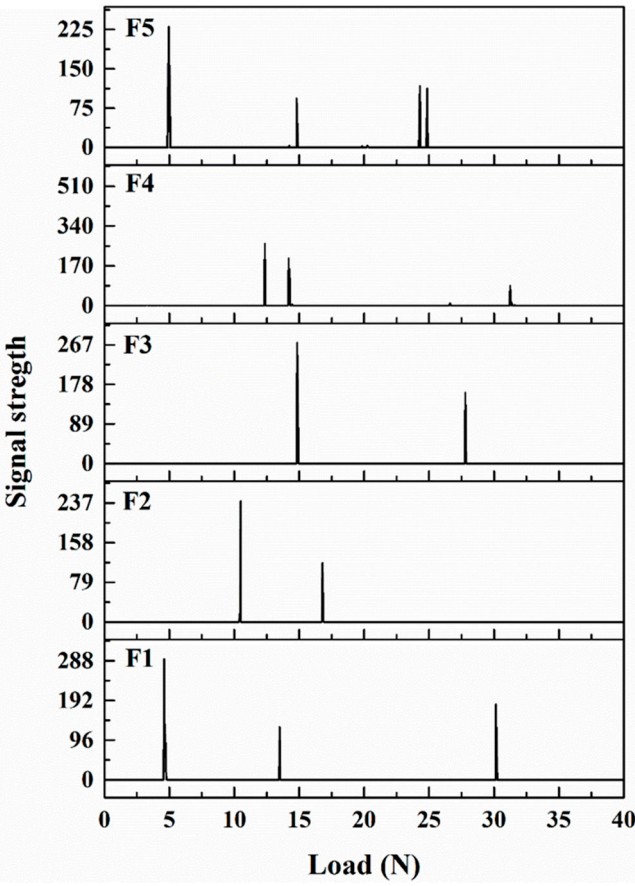

**Figure 6.** The bonding force of MAO coatings under different negative voltages.

**Table 5.** The bonding force and load strength of MAO coatings under different negative voltages.

|  | F1 | F2 | F3 | F4 | F5 |
|---|---|---|---|---|---|
| Bonding force (N) | $4.60 \pm 0.14$ | $10.45 \pm 0.31$ | $14.85 \pm 0.44$ | $12.35 \pm 0.37$ | $4.95 \pm 0.15$ |
| Bonding strength (MPa) | $36.6 \pm 1.10$ | $83.2 \pm 2.49$ | $118.2 \pm 3.54$ | $98.3 \pm 2.94$ | $39.4 \pm 1.18$ |

### 3.7. Potentiometric Polarization Test

Figure 7 shows the polarization curve of MAO coating samples in 3.5 wt.% NaCl solution by potentiodynamic polarization tests. The self-corrosion potential ($E_{corr}$) of the substrate (Table 6) is $-0.713$ V with a corrosion current density ($I_{corr}$) of 161 μA/cm$^2$. After the MAO operation, the sample self-corrosion potential and polarization resistance ($R_p$) increase significantly, while the $I_{corr}$ is significantly reduced. As the negative voltage increases, the self-corrosion potential rises from $-0.583$ V in F1 to $-0.488$ V in F3 and then decreases to $-0.539$ V in F5. The F3 coating also has the lowest corrosion current density and highest $R_p$, 3.64 μA/cm$^2$ and 7.0 kΩ·cm$^2$, respectively, implying that the F3 coating has the best corrosion resistance among all samples that may be attributable to the following reasons: firstly, the high content of α-Al$_2$O$_3$ provides great insulation performance and contributes to slowing down the coating electrochemical corrosion; secondly, the F3 coating, with more pancake-like morphologies and tiny holes produced by B-type discharge, is more compact and flat, which can effectively protect the substrate from NaCl solution.

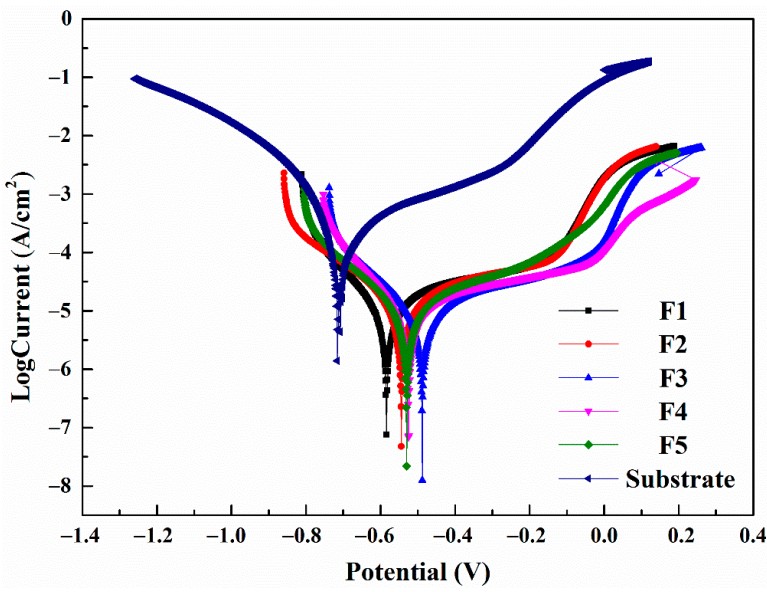

**Figure 7.** Polarization curves of coatings under different negative voltages.

**Table 6.** Fitting parameters by Tafel polarization curves of samples.

|  | $-E_{corr}$ (V) | $I_{corr}$ ($\mu A \cdot cm^{-2}$) | $\beta_a$ (V dec) | $-\beta_c$ (V dec) | $R_p$ ($k\Omega \cdot cm^2$) |
|---|---|---|---|---|---|
| Substrate | 0.713 | 161 | 0.296 | 0.095 | 0.1 |
| F1 | 0.583 | 4.81 | 0.134 | 0.099 | 5.1 |
| F2 | 0.544 | 4.54 | 0.117 | 0.095 | 5.0 |
| F3 | 0.488 | 3.64 | 0.103 | 0.138 | 7.0 |
| F4 | 0.524 | 4.79 | 0.110 | 0.167 | 6.0 |
| F5 | 0.539 | 4.92 | 0.140 | 0.103 | 5.2 |

*3.8. Neutral Salt Spray Experiment*

In Figure 8, the 10B21 steel has the worst corrosion resistance. After 12 h, severe corrosion appears on the surface, and FeCl$_3$, with a typical rust-red color, is generated on the sample surface. As the testing time is prolonged to 48 and 72 h, the surface features corroded more severely large black areas, rusted red spots, and deep corrosion pits showing up on the substrate surface. The corrosion rate is 6.9 g·m$^{-2}$·h$^{-1}$, as shown in Figure 9. Because of the coatings, the corrosion degree of F1 and F3 sample declines compared with that of the substrate. Specifically, the F1 coating is porous with plenty of iron oxides and is able to generate FeCl$_3$ in the NaCl solution at the early stage of salt spray. After 12 h, uneven rust red begins to show up on the surface. With time prolonged to 72 h, the continuous infiltration of NaCl solution from the holes on the coating leads to a gradual increase on the rusted areas of the surface. The surface of F1 coating is protected from a bad corrosion with a relatively low corrosion rate of 3.7 g·m$^{-2}$·h$^{-1}$. For F3, a few tiny holes are eroded after 12 h. The corrosion spots are aggravated with time, but the integrity of the F3 coating basically remains. Compared with those of F1, the corrosion degree and speed of F3 are lower as with a corrosion rate falling down to 2.26 g·m$^{-2}$·h$^{-1}$ (Figure 9) after 72 h, 2/3 lower than that of the substrate. The test also illustrates that the F3 coating under −100 V exhibits better corrosion resistance in that it can effectively prevent NaCl solution from entering the interior of the coating and reacting with the steel substrate.

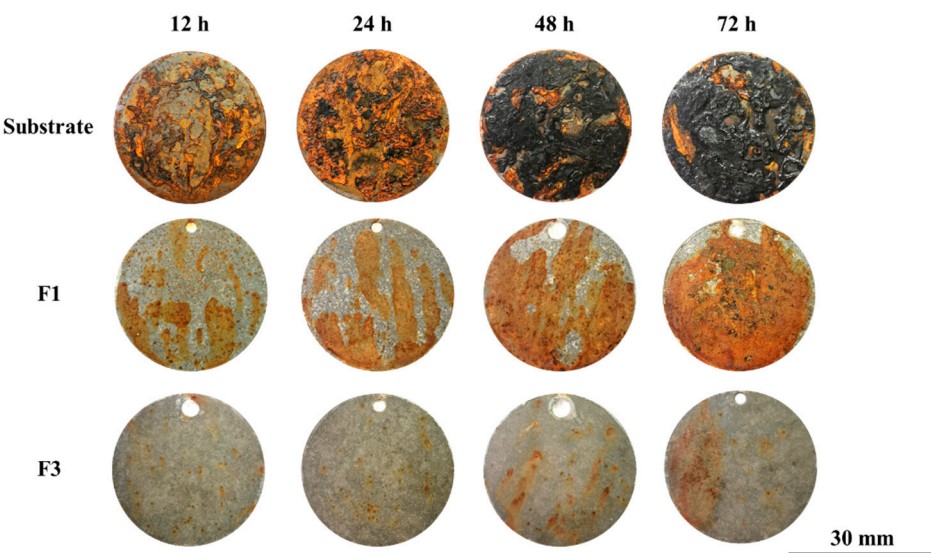

**Figure 8.** Salt spray corrosion of the MAO coating at different times under different negative voltages.

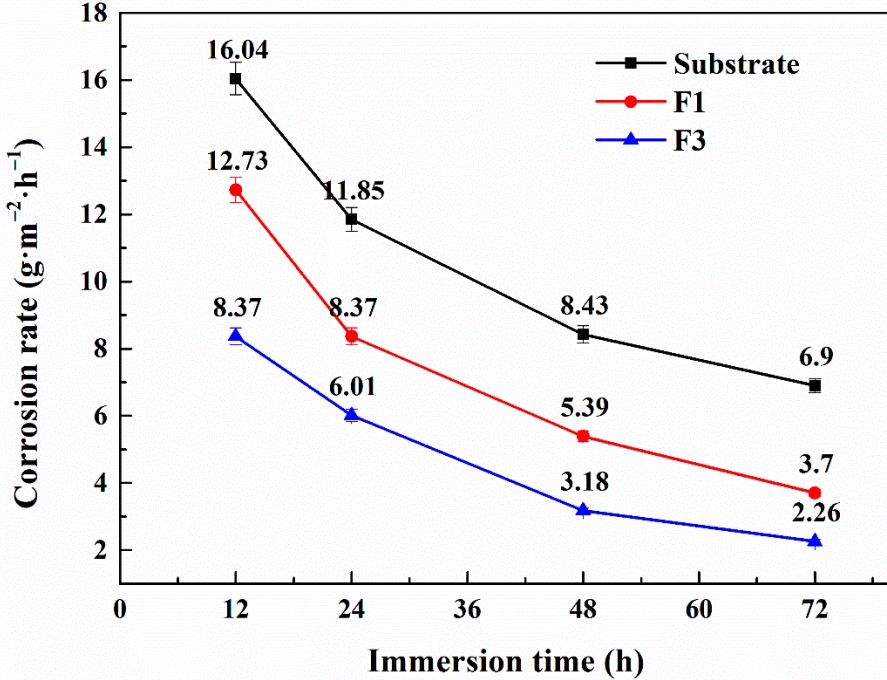

**Figure 9.** Corrosion rate curves of MAO coatings under different negative voltages.

## 4. Conclusions

A microarc oxidation (MAO) coating was successfully prepared on the surface of 10B21 steel in the electrolyte of aluminate and phosphate. The influence of negative voltage on the microstructure morphology, bonding strength, and corrosion resistance of the coatings was studied, and the following conclusions were reached:

(1)  All MAO coatings are composed mainly of $\alpha$-$Al_2O_3$, $\gamma$-$Al_2O_3$, $Fe_3O_4$, and $FePO_4$, and the negative voltage can significantly affect the surface morphology and composition in the coating;

(2)  The coating without a negative voltage or with a lower negative voltage has a higher content of $\gamma$-$Al_2O_3$ and a porous and loose structure. Under a negative voltage of $-100$ V, the content of $\alpha$-$Al_2O_3$ in the coating is increased, and the surface morphology

is more compact and denser with a porosity of 7.4%. The coating thickness rises to 34 μm, and the bonding strength reaches 118.2 $\pm$ 3.54 MPa;

(3)  The appropriate negative voltage also greatly improves the corrosion resistance of the coating, and after 72 h of salt spray testing, the corrosion rate of the sample declines to 2.26 g·m$^{-2}$·h$^{-1}$. The coating obtained under $-100$ V exhibits the best corrosion resistance, significantly improving the corrosion resistance of 10B21 steel.

**Author Contributions:** M.X.: Writing—original draft, formal analysis. T.L.: Methodology, visualization, formal analysis. Y.Z.: Writing—review and editing, supervision. M.C.: Writing—review and editing, supervision, project administration. All authors have read and agreed to the published version of the manuscript.

**Funding:** The authors acknowledge the financial support for this work from the Key projects of the Joint Foundation of the National Natural Science Foundation of China (U1764254), and National Nature Science Foundation of China (51871166 and 51801137).

**Institutional Review Board Statement:** Not applicable.

**Informed Consent Statement:** Not applicable.

**Data Availability Statement:** Not applicable.

**Conflicts of Interest:** The authors declare that they have no known competing financial interests or personal relationships that could have appeared to influence the work reported in this paper.

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
