# Peer review of "The Influence of Negative Voltage on Corrosion Behavior of Ceramic Coatings Prepared by MAO Treatment on Steel"

_coatings, doi:10.3390/coatings12050710_

Round 1

Reviewer 1 Report

The submitted article is devoted to the study of the influence of negative voltage on corrosion behavior of ceramic coatings prepared by MAO treatment on steel.

There are several significant comments on this work.

1) The abstract is poorly structured. It needs style improvement.

2) The submitted manuscript is very close to reference [21] (electrolyte, substrate). However, the authors do not compare the obtained results with those known in the literature.

3) Specify, please, the mechanism of formation of coatings after applying a negative voltage. All the above reactions for the formation of aluminum oxide are anodic. How does the application of negative potentials affect electrode reactions?

4) The XRD patterns of the obtained samples are very similar. What is the reproducibility of the obtained results? If only single samples are studied, then the conclusions of the study are invalid.

5) How thermodynamically substantiated are the conclusions of the authors on the effect of negative voltage on the composition of the formed coatings?

6) The conclusion of the authors about the optimal voltage of -100 V is not justified. All results are very close, no reproducibility data, error values are not given.

Reviewer 2 Report

I can recommend the publication of the manuscript after the following major comments:

Insert references for all mathematical formulas.

Figs. 2 and 3: insert in-text information about the SEM micrographs, such as: magnification, acceleration voltage, working distance, and image resolution pixels.

Page 3, line 124: insert explanations for all parameters of eq. (1).

Pages 5 (line 177) and page 6 (line 186) - insert more details about “"....produces a rough surface...", “.... surface roughness...”. Please, specify the name of the roughness parameter. If possible, insert some height parameters for surface texture, such as a) Height parameters: Root mean square height Sq [nm]; Skewness Ssk [-]; Kurtosis Sku [-]; Maximum peak height Sp [nm]; Maximum pit height Sv [nm]; Maximum height Sz [nm]; Arithmetic mean height Sa [nm]. b) Surface texture directions: First direction [°]; Second direction [°]; Third direction [°]; Isotropy [%].

Page 9, fig. 4: try to use a uniform font text in pictures.

Page 9: line 259 - insert more details about “....the surface of the coating becomes smoother,..”. Explain what statistical surface parameters were applied to support this statement.

Page 10: line 266 - insert more details about “...produces much heat...”. “...leading to a rapid rise....”.

Page 11 lines 285, 287, 293 - insert more details.

Page 12: line 323 - insert more details about “....is more compact and flat,..”. Explain what statistical surface parameters were applied to support this statement.

Page 13, fig. 8: try to use a uniform font text in pictures.

Insert more details about the statistical analysis applied (methods, software, and so on).

References are not written according to the Guide of Authors (ref. [14], [38], and so on).
Insert Data availability statement.
If possible, I recommend this reference be cited:
[1] Țălu, Ș. (2015). Micro and nanoscale characterization of three dimensional surfaces. Basics and applications. Napoca Star Publishing House: Cluj-Napoca, Romania (pp. 21-27).
This paper can be published after the mentioned revisions.

Reviewer 3 Report

In this work, the properties and corrosion resistance performance of 10B21 after coating was investigated. Five different coatings were prepared by micro-arc oxidation (MAO) under different negative voltages in aluminate and phosphate electrolytes. Then , the influence of negative voltages on the composition, microstructure, bonding strength, and corrosion performance of these MAO coating was systematically studied. A model proposing the mechanism of coating growth under different negative voltages on 10B21 steel was also proposed. This is generally good incremental work with a lot of signifiant parameters derived from analysis which allowed to select optimal oxidation conditions for the coating in terms of its corrosion protection role. The analysis itself is done with sufficient scientific care. Therefore, in general I think that this article can be accepted for publication with a minor recommendation:

- please check carefully English style and grammar. Particularly Abstract and Conclusions need to be better written to not discourage potential readers to appreciate substantive aspects of the work.

Round 2

Reviewer 1 Report

The authors have addressed majority of comments of the reviewers and revised the manuscript accordingly. I recommend accepting the manuscript  for publication. 

Reviewer 2 Report

Accept in present form.